# Should I Stay or Should I Go? A Qualitative Exploration of Stigma and Other Factors Influencing Opioid Agonist Treatment Journeys

**DOI:** 10.3390/ijerph20021526

**Published:** 2023-01-14

**Authors:** Victoria Rice Carlisle, Olivia M. Maynard, Darren Bagnall, Matthew Hickman, Jon Shorrock, Kyla Thomas, Joanna Kesten

**Affiliations:** 1Population Health Sciences, Bristol Medical School, University of Bristol, Bristol BS8 1TL, UK; 2School of Psychological Sciences, University of Bristol, Bristol BS8 1TU, UK; 3Avon & Wiltshire NHS Mental Health Trust, Specialist Drug and Alcohol Services, Colston Fort, Montague Place, Bristol BS6 5UB, UK; 4The National Institute for Health and Care Applied Research Collaboration West (NIHR ARC West) at University Hospitals Bristol and Weston NHS Foundation Trust, Population Health Sciences, Bristol Medical School, University of Bristol, Bristol BS8 1TL, UK; 5The National Institute for Health and Care Health Protection Research Unit (HPRU) in Behavioural Science and Evaluation, University of Bristol, Bristol BS8 1TL, UK

**Keywords:** opioid agonist treatment, methadone, stigma, socioecological model, drug treatment, qualitative

## Abstract

**Highlights:**

**Abstract:**

(1) The harm-reduction benefits of opioid agonist treatment (OAT) are well-established; however, the UK government’s emphasis on “recovery” may be contributing to a high proportion of people leaving treatment and low retention rates. We wanted to develop a rich and nuanced understanding of the factors that might influence the treatment journeys of people who use OAT. (2) We explored factors at each level of the socioecological system and considered the ways these interact to influence treatment journeys in OAT. We carried out semi-structured interviews with people who use OAT (n = 12) and service providers (n = 13) and analysed data using reflexive thematic analysis. (3) We developed three themes representing participant perceptions of treatment journeys in OAT. These were: (1) The System is Broken; (2) Power Struggles; and (3) Filling the Void. (4) Conclusions: The data suggest that prioritisation of treatment retention is important to preserve the harm-reduction benefits of OAT. Stigma is a systemic issue which presents multiple barriers to people who use OAT living fulfilling lives. There is an urgent need to develop targeted interventions to address stigma towards people who use OAT.

## 1. Introduction

Opioid agonist treatment (OAT) is the first-line, evidence-based treatment for individuals seeking help with opioid dependency worldwide [1]. Treatment is based on the provision of the medications methadone or buprenorphine, alongside psychosocial components, such as contingency management or motivational interviewing [2]. In the UK, medications used in OAT are generally collected from community pharmacies. In early treatment, guidelines recommend that consumption of medications is supervised by a pharmacist (“supervised consumption”: [3]). OAT medications have a longer half-life than heroin, meaning their effects last at least 24–36 h. By removing the need to frequently obtain and use heroin, the overarching aim of OAT is to reduce and eliminate heroin use and provide individuals with the time and stability to tackle the psychosocial issues underlying their dependency [4]. According to current UK government guidelines [5], OAT medications should not be lowered unless side effects are experienced, or the current dose is not working well. Further, people who use OAT should not be encouraged to reduce their medications gradually over the course of treatment; rather, detoxification should only take place if and when people are ready to do so. Detoxification that takes place over a period of around twelve weeks in the community or 28 days as an inpatient is considered to be safe [5].

As a result of the associated reductions in injecting drug use, OAT is effective in lowering the transmission of bloodborne viruses, such as HIV and hepatitis C [6,7,8,9], and reducing the risk of overdose [10]. Retention in OAT is also associated with fewer hospital admissions for injecting-related infections [11]. However, these benefits may be undermined by poor treatment retention and relapse to illicit opioid use. Retention rates in OAT are around 57 percent at twelve months and just 38 percent at three years according to a recent systematic review of 67 studies [12]. Unless individuals die, go to prison or are transferred to another area, treatment is ended in one of two ways: either by “treatment completion” (planned discharge) or by “dropping out” (unplanned discharge). According to the UK Department of Health and Social Care [2], the aim of OAT is for individuals to progress from maintenance to detoxification and eventually to abstinence; however, in opioid treatment, the most common reason for leaving is as a result of dropping out, with only 25% completing treatment [13]. This is reflected locally in Bristol, where our previous work identified that those individuals that drop out of treatment were also more likely to re-present to treatment in the future [14]. Retention, therefore, is a key consideration in OAT as “cycling in and out of treatment” [15] exposes individuals to greater risk than when retained in treatment long term. However, vulnerabilities at the start and immediately after treatment are different for methadone and buprenorphine, with the early weeks of treatment being a particularly vulnerable time for people who use methadone (but not buprenorphine) with an elevated risk of all-cause mortality and drug-related poisoning, whereas the four weeks following discharge are associated with greater risks for both types of OAT [16].

Despite the clear harm-reduction benefits of retention of OAT programmes, the UK government’s drug strategy at the time the current study was conducted [17] only briefly mentions harm-reduction initiatives, including OAT, which is concerning in the context of record levels of opioid deaths in this country [18]. Instead, a focus is placed on helping more people to “recover” from drug dependency. However, the word “recovery” is conflated with abstinence or “treatment completion”. This is problematic as it may pressurise individuals to detox from OAT medications before they are ready, ultimately resulting in a return to illicit opioid use [19]. The focus of OAT in the UK has shifted from long-term maintenance based on harm-reduction principles towards treatment providers being measured on the numbers of service users successfully “completing treatment” [20,21]. Additionally, the conceptualisation of recovery as a linear process with a binary outcome of success versus failure does not fit with the scientific view of addiction as a chronic, relapsing disorder requiring ongoing management rather than “cure” [22]. Finally, there is a high level of consensus amongst people who use drugs that recovery is more than simply an absence of substances but rather something that involves more holistic improvements to relationships, housing, health, emotional stability, employment and community re-integration [23,24,25]. Crucially, all these outcomes can be achieved within long-term OAT (MAR: medication-assisted recovery) and do not require detox from medications.

One factor that appears to influence OAT journeys is stigma [26,27,28,29,30,31]. Stigma can be thought of as an “attribute that is deeply discrediting” [32] or “a set of undesirable characteristics” [33]. A useful framework considers stigma as three separate mechanisms: anticipated stigma (an expectation of being judged or stereotyped); enacted stigma (that which has been directly experienced); and internalised stigma (the direction of negative stereotypes and perceptions towards oneself) [34,35].

There is a breadth of qualitative literature exploring experiences of OAT; however, these tend to focus on factors in isolation, for instance, fear [36]; social support [37]; or stigma [26,27,28,29,30,31]. Socioecological approaches allow for the exploration of issues across five levels: individual, intrapersonal, organisational, community and policy [38]. Such models are increasingly used to study public health issues, including opioid use and dependency (e.g., [39,40,41,42,43,44]). However, only a small number of studies have adopted this approach to specifically explore experiences of OAT [39,45,46], and, to our knowledge, no studies have been conducted in a UK treatment context. Recently, Dame Carol Black’s review of drugs [47,48] identified issues of chronic underfunding and called for increased funding for both drug treatment and wider recovery support, as well as the need to take a systemic approach to addressing drug use. As a result, the UK government updated its national drugs strategy [49] and is investing GBP 80 million in treatment and recovery services [50]. Similarly, President Biden has recently announced support of USD 1.5 billion to address the opioid crisis in the United States [51]. Given that people seeking help for opioid use make up the largest proportion of the treatment population [13], this is therefore a timely and relevant opportunity to provide evidence of the systemic factors that impact on experiences of OAT.

In the current study, we sought to gain a rich understanding of the factors that may influence retention, completion and treatment experiences within OAT in Bristol, England. Informed by socioecological models, we sought to understand the way different socioecological aspects interact to influence treatment outcomes in OAT.

## 2. Material and Methods

### 2.1. Study Design and Setting

We carried out a qualitative study within third sector and NHS drug treatment providers in Bristol. Bristol is a medium-sized city in the South West of England with a population of just under 500,000 and an estimated 4130 opioid users [52]. Participants included people currently or previously using OAT (methadone/buprenorphine) and staff working with them. OAT in Bristol is delivered by “shared care workers” in an arrangement between GP (general practice) surgeries and a third-sector organisation. This means that people who use OAT are seen by specialist drug and alcohol treatment staff at their own GP surgery. The shared care worker has responsibility for discussing prescriptions of medications (“scripts”) as well as delivering psychosocial aspects of treatment. In preparation for this study, the first author undertook a process of familiarisation of these services, including the shadowing of shared care workers and a specialist addiction clinical psychologist. This allowed for a sensitivity to the needs and experiences of both service users and staff involved with the local delivery of OAT and aided in rapport building with participants.

We developed all documentation relating to recruitment, including posters, information sheets and consent forms, in collaboration with an individual with lived experience of OAT, who has previous experience of contributing to similar research studies. This enhanced reflexivity by providing an alternative perspective on the topic, for instance, the inclusion of language in the participant information sheet that would be familiar to and understandable by people who use OAT. We also worked closely with drug treatment staff, who made valuable contributions to the research process, for instance, by highlighting potential barriers to recruitment.

### 2.2. Sampling and Recruitment

We adopted a purposive sampling strategy to recruit adults currently or previously using OAT from a diverse range of service users at different stages of treatment, and we aimed to recruit a balance of male to female participants that was roughly reflective of the local treatment population as a whole (28% female). Those previously using OAT were recruited via Community Discharge Link workers and had recently been discharged from treatment. The Community Discharge Link (CDL) team was a recent initiative in Bristol at the time of the study set up with the aim of supporting people who are detoxing from OAT medications in the community (as opposed to residential detox/rehabilitation). Once a planned detox has been agreed between a person who uses OAT and their shared care worker, the CDL worker provides additional psychosocial support throughout the tapering and detox period. Current OAT service users were recruited via six shared care workers based at different GP surgeries across the city. Shared care workers explained the study to potential participants and passed on their contact details to the researcher if they were interested in participating. We also wanted to recruit a selection of service providers from across the drug treatment network locally, with the aim of developing a rich and nuanced understanding of treatment journeys within OAT from multiple perspectives. These were recruited via staff email, posters on staff notice boards and after a presentation by the lead author at a staff meeting.

We determined sample size according to the concept of Information Power [53]. This approach states that fewer participants for qualitative studies are required if a sample holds more information directly relevant to the research questions of the study. Guided by this principle, we planned for a moderate-to-large sample size of around twenty-four participants (twelve service users, twelve service providers), with regular reviews of the appropriateness of this throughout data collection. After conducting twenty-five interviews, we confirmed that our sample was more than sufficient to meet Information Power considerations.

We developed separate topic guides for service users and service providers; however, the broad aim was the same for both groups—to gain a rich understanding of OAT treatment journeys. Topic guides were used flexibly to ensure that emergent topics could arise during the interview process and to allow participants to be the experts of their own experiences. As a result, topic guides were developed iteratively throughout the data collection phase as it became apparent which topics were most salient.

Face to face interviews took place in a private room at the third-sector drug treatment centre. Participants were additionally offered the option of a telephone interview if more convenient; however, all preferred to be interviewed in person. Interviews took place between October 2019 and February 2020 and lasted between 31 and 87 min. We obtained written, informed consent for all participants prior to the interviews, which were recorded on an encrypted audio-recording device. Service users were reimbursed GBP 10 cash for their time. Service providers participated during their normal working hours and therefore were not financially reimbursed for their time. The study was approved by the NHS North East, York Research Ethics committee (reference number 18/NE/0242).

### 2.3. Data Analysis

We analysed interview data using reflexive thematic analysis [54,55], as it allowed us to combine an inductive approach to coding, with the ability to later consider themes within an existing analytical framework (the socioecological model). Consistent with the applied nature of our research aims, we adopted a more experiential and descriptive mode of analysis to make sense of individuals’ experiences of OAT. This was situated within a critical realist orientation [56,57].

Throughout data collection and analysis, the first author kept a research journal. This was particularly helpful in enabling reflection on the research process and interpretation of the data. She debriefed with another member of the research team after every three interviews; this allowed for further reflection and also an opportunity to seek advice about any challenging issues. We analysed service user and service provider transcripts as a single data corpus but highlighted similarities and differences between the two groups when they were apparent in the data. Analysis involved a process of reflection on both the data as well as the perspective of the researchers. Analysis was led by the first author in collaboration with two other authors (J.K. and O.M.), none of whom report lived experience of using opioids or OAT. V.C. has previously volunteered in the drug and alcohol treatment sector as well as having lived and familial experience of (non-opioid) addiction. This analysis was supported by the wider study team and an individual with lived experience of using opioids and OAT.

Interviews were transcribed, and the first author listened back to each interview several times, firstly to check accuracy and anonymise transcripts and secondly to immerse herself fully in the data. The first author conducted initial coding directly onto transcripts, making notes on each of the interviews and complementing the research journal kept throughout data collection. Coding was inductive in nature with no aim to fit the data into existing theories. The first author then imported all transcripts into QSR NVivo 12 software [58].

After the first author coded all relevant sections of transcripts that were related to our research questions, she had several hundred codes. She then clustered conceptually similar codes together to form candidate themes. We defined a theme as a concept that *“captures something important about the data in relation to the research question and represents some level of patterned response or meaning within the dataset*” [59]. Titles of themes were chosen to reflect the “central organising concept” of that theme. 

The first author then produced theme summaries for each of the candidate themes, and these were discussed with two other members of the team (J.K. and O.M.) before finalising the themes. This allowed us to define the boundaries of each theme and consider relationships between them. During this process, we endeavoured to consciously consider all other possible explanations for the data; this involved interrogating our own assumptions and experiences and trying to take an alternative perspective. Finally, our themes were considered in relation to the socioecological model [38] to add analytical depth to the analysis and explore the way that individual, inter-personal, organisation, community and policy factors interact to influence experiences of OAT.

## 3. Results

Characteristics of participants are shown in Table 1 and Table 2. Given the small number of service provider participants and the likelihood that they could be identified by their job titles, in the table we have presented characteristics for service providers without pseudonyms (however, these are provided after data excerpts). Pseudonyms for service user participants are provided in Table 1. Service user data extracts are indicated by “SU” and service providers by “SP”.

We developed three themes in relation to our participants’ experiences and perceptions of treatment journeys in OAT in Bristol: (1) The System is Broken; (2) Power Struggles; and (3) Filling the Void. An overview of the findings in relation to the socioecological model is shown in Figure 1.

### 3.1. The System Is Broken

The first theme explores interactions between the policy and organisational level via the influence of drug policy and the way that this impacts on how treatment is delivered and received. Perhaps naturally, service providers contributed more heavily to this theme; however, discussion of these issues did take place with several of the service users too.

The notion of “the system” as a constraining force was commonly evoked, “*The boxes have got to be ticked because there wasn’t any funding and it’s not really anyone’s fault, this is just the system and its… you know, now it just feels so different*” (Anne-Marie, SU). One facet of systemic constraints, driven by an increase in caseloads and reduced funding, was the sense of time pressures. This concept was often referred to in relation to discussions of psychosocial aspects of treatment: “*you’ve got half-an-hour once a fortnight to delve into why somebody is using drugs*” (Gabor, SP). At the inter-personal level, time pressures appear to impact on trust building between service provider and service user:


*I think another barrier [to recovery] is time…often I don’t have time to build those working relationships with people within the capacity of our sessions, or a day, or a surgery, or a timetable. (Russel, SP).*


Service providers described experiencing pressure to increase the number of drug-free discharges (as the result of changes to drug policy) whilst at the same time not being provided enough time to work in depth with service users to address the reasons for their drug use:


*I’ve got to get this person drug free, how am I going to get them drug free? You’re not, that’s not going to happen with half an hour, once a fortnight (Gabor, SP).*



*The pressure…Give us time to do our jobs! (Eric, SP).*


The implications of this are that there is minimal scope for service providers to assist people in increasing their stability outside of that which is offered by OAT medications.

The perception of the system as a constraining force resulted in a sense of tension between policy-level ideologies and the realities of delivering client-centred care at the organisational level, “*They talk about recovery a lot in the recovery agenda but actually it’s just leaving treatment isn’t it? there’s not really anything about what happens before that*” (Marc, SP).

Such recovery narratives, which are based on the assumption that recovery is equal to abstinence from all opioids, including OAT, were perceived to be damaging, as they imply that long-term OAT is problematic, which may encourage people to detox before they are ready. For instance, service user Jack explains, “*It helps control the cravings…but I have to get off the buprenorphine now”*. When asked whether this was something Jack desired, concerningly he replied “*No, I don’t want to at all but like obviously they’re [staff] going to want me to”.* This illustrates the way that internalising recovery narratives contributes to reducing service users’ sense of autonomy, disempowering and disconnecting them from steering their own recovery (discussed under “Power Struggles”, below). Further, viewing recovery as an endpoint rather than a self-defined process disregards the fact that people may be able to complete treatment drug-free; however, such “recoveries” may be unsustainable without further support, “*A lot of the recoveries that we participate in are quite thin, it’s people coming off scripts [prescription for OAT medication] but holding on by the seat of their pants”* (Ronnie, SP). *“Holding on by the seat of their pants”*, therefore, is a fitting metaphor for the instability associated with a premature detox.

Another issue identified by service providers was the frequent recommissioning of services by the local authority. The most recent round of recommissioning in Bristol meant that services that were previously provided by one organisation had become split between two providers. Eric, a service provider, highlights the impact this may have on retention as waiting lists for group work have increased and *“windows of opportunity”,* the period of time in which to engage service users, narrowed to the point where people drop out and are left vulnerable as *“there is nothing there for them”.* As he says poignantly, *“it’s heart-breaking to watch”.*

The perception of treatment journeys as a complex and circuitous process was evoked by several participants, “*…within four months I was trying to stop and within six months I was within rehab and it’s been like that ever since”* (Peter, service user). Our data suggest that the current treatment system does not appear to adequately address the inherent complexity of addiction. In Bristol, such complexity may be increasing as a result of service recommissioning and reductions in treatment budgets, which have resulted in shared care seeing more complex and chronic presentations (for instance, those with additional mental health needs that were previously dealt with by the National Health Service). This may mean increasing numbers of clients who are not able to currently work towards detox, “*[…] you’ve got a hardening, more chronic case load; we’ve also got the more complex clients we never used to see before”* (Jenny, SP). This could potentially pose two challenges for treatment organisations; either more people are retained in treatment as they are more complex and not able to detox safely (increasing caseloads), or they are so complex that they are at increased risk of dropping out and relapsing. Ultimately, a system which fails to accommodate the complexity of addiction and recovery is likely to only facilitate limited, short-term improvements to the lives of people who use opioids:


*When they stop using opioids, hey it’s a successful completion. Are they? Or are they now just angry young men in an old man’s body who we are not supporting to actually grow emotionally and grow as people…do they just get left by the wayside because they’re no longer opioid users? […] what have we left them with? And we wonder why people relapse (Brené, SP).*


The concept of “*what have we left them with*” that Brené describes above is important as it relates to “voids” that are created when people stop using opioids and detox from OAT (discussed under the theme “Filling the Void”, below).

Finally, relationships between organisations in the treatment system may impact on experiences of OAT. For instance, where GPs are directly involved in OAT (as they are in Bristol), support may vary between surgeries and GPs:


*… it varies across the city and it depends on how responsive the GPs are willing to be in that scenario, so if somebody’s missed three days [picking up medications] and they approach the surgeries directly on day 4, some GPs are really proactive, others will say, ‘No, you need to talk to [treatment organisation], we don’t want anything to do with that’ (Ronnie, SP).*


This may mean that the level of support offered to service users is inconsistent between GP surgeries, potentially widening health inequalities. It is also possible that the advantages associated with delivering OAT from GP surgeries rather than in the drug treatment setting (e.g., they are typically local to the service user, and discussions with GPs about scripting and other issues are readily available) may be undermined by challenges such as the struggle that service users have with adhering to the structure and restrictions of being seen through the GP appointment system. This may be particularly challenging for individuals experiencing less stability, such as those who are homeless or using opioids illicitly.

#### Re-Framing Recovery

Table 3 shows a broad range of possible outcomes, suggested by our participants, that may indicate recovery—a concept that is re-framed here as a multi-faceted and highly individual process which is in opposition to the “recovery as abstinence” narrative that is perpetuated by UK government drugs policies. In line with previous research [23,24,25], these outcomes suggest a more holistic view of recovery, one which encompasses physical and social outcomes and gives space for both harm reduction and abstinence.

### 3.2. Power Struggles

This theme relates to issues of power and the way in which service users are constrained by power imbalances that exist in relation to their treatment. Patterning in this theme coalesced around two sub-themes of “autonomy” (or lack thereof) and “stigma”, where powerlessness provides the ideal conditions for stigma to thrive [28,33].

#### 3.2.1. Autonomy

Both service users and providers recognised the inherent power imbalance between staff, who hold *“the power of the blue script”* (Jenny, SP), and service users, who are required to conform to the multiple expectations of treatment, *“We introduced things like if you don’t come to your appointments, we’re going to reduce your script and kick you off it*” (Brené, SP). The fact that service providers hold the dual role of “gatekeeper” of medications as well as “counsellor” poses a barrier to trust building and may impact on therapeutic alliances in OAT [60]. Whilst such power imbalances may be difficult to eradicate entirely, due to the need to balance service users’ wishes with the reduction of risk, it is likely that increasing service users’ sense of autonomy may mean a more positive and constructive experience of treatment. One suggested method of increasing autonomy might be by allowing for less frequent appointments for those in long-term stable treatment:


*[…] it’s always a fear in the back of your head [losing access to OAT] so yeah you feel like it’s a big job to hold a script…it’s quite controversial I suppose isn’t it to say it….it’s not like give it to them for 10 years…it could be every three months or six months to have a chat…(Anne-Marie, SU).*


Anne-Marie goes on to explain that she attempts to regain a sense of autonomy by “pushing back” against the system, “‘You’ve got to cut down [buprenorphine], you don’t have a choice’…I’m very stubborn so if someone’s trying to push me to stop I’m pushing back”. Such “pushing back” may undermine harm reduction aims of treatment by forcing people to conceal their drug use for fear of losing their script.

Service users reported a range of experiences where they perceived their autonomy and agency to be limited within and by treatment. This included frustration with restrictions on their freedom to be able to go away due to daily collections of medications over long periods; of feeling bound by the addictive properties of methadone (“liquid handcuffs”, e.g., [61]); feeling pressure from others to detox; or feeling pressurised by a treatment system that incentivises drug-free discharges (discussed above). This mismatch between service provider and service user motivations can result in the latter feeling that their recovery is being “*steered”* by others, reducing their feelings of autonomy over their treatment:


*… I was just chasing it [recovery] and then it felt like it was other people’s plan […] so it was like you were sort of more steered by other people rather than it being ‘right, I want to do this and then I want to do that and do that’ (Keith, SU).*


#### 3.2.2. Stigma

Stigma can be thought of as an *“attribute that is deeply discrediting*” [32] or *“a set of undesirable characteristics”* [33]. A useful framework considers stigma as three separate mechanisms: anticipated stigma (an expectation of being judged or stereotyped); enacted stigma (that which has been directly experienced); and internalised stigma (the direction of negative stereotypes and perceptions towards oneself) [34,35].

Experiences of stigma, possibly *“the greatest barrier to people’s recovery”* (Gareth, SP), were discussed either explicitly or alluded to throughout the dataset. It was perceived to be present in all areas of service users’ lives, from interactions with friends and family; in GP surgeries, pharmacies and other healthcare settings; and as a result of stigmatising policies:


*My Dad made a comment [to participant] about some druggies getting their medication before he got his or something […] (Ringo, SU)*



*It might mean that the stigma [they] receive as [they] walk into a GP’s surgery—often even the people behind reception, [they] could have been at school with, or they’re a family friend or something (Stephen, SP)*



*When you’ve got treatment systems that are authoritarian and judgemental and massive […] Part of this is increasingly driven by the whole recovery definition so recovery is about not using drug and alcohol so by definition if you are using drugs and alcohol you’re not a success so I think that helps drive that [stigma] (Brené, SP).*


Our data suggest that the pharmacy is a particularly strong source of enacted and anticipated stigma, coming from both staff and customers, *“[pharmacy staff] treat us like we’re sub-human”* (Peter, SU). Discussions of the pharmacy setting often alluded to the “othering” of people who use OAT. For instance, pharmacists restricting times that service users can collect medications or serving other customers first:


*… they’ve got the queue there for people with normal prescriptions and then they’ve got you on the side so you’re like palmed off to the side yeah and they will serve every person in that queue before they’d even come to you man. (Ringo, SU).*


Eric, a service provider, highlights that whilst collecting medications from a local pharmacy is more convenient, it may expose service users to stigma, possibly due to the fact that the pharmacy is the environment where people who use OAT are most likely to interact with members of the wider community, exposing them to the *“public gaze”* [28]: “*If it is in a local pharmacy, there will probably be family members in there. They will be taking you to the consultation room and everyone knows what the consultation room is for”.* Eric goes on to use the words “*a badge of shame*”, which poignantly captures the emotional impact that internalised stigma has on service users. Above, Ringo uses the phrase *“normal prescriptions”*, suggesting the internalisation of discourses of normality and morality that surround drug use and OAT. Internalised stigma and the resulting shame therefore present an additional barrier to recovery and retention to overcome:


*It’s difficult and it does make you feel less than and not worthy and you know, you’re already beating yourself up because you’re in the situation you’re in so you don’t need someone looking down on you […] (Keith, SU).*


For service users collecting their medications daily, the pharmacist is likely to be the professional that they have the most contact with. Such seemingly insignificant steps such as using someone’s name and *“treating [them] the same as the next person*”, as Keith’s pharmacist does, have the potential to have a profound effect on the life and treatment experiences of people who use OAT. Pharmacists therefore have the potential to be a key source of social support for people who use OAT (see [62] for a discussion).

### 3.3. Filling the Void

Previous work has used the phrase *“filling the void”* [63] to describe the absence left behind after starting OAT and no longer using heroin. Although some of our participants continued to use heroin, the concept of voids was evoked by several participants in our study. We identified two major voids in this respect—time/purpose and social connectedness.

#### 3.3.1. A Sense of Purpose: Filling the Time Void

Obtaining and using heroin is a time-consuming activity, which punctuates the daily lives of people who use heroin; with the cessation of illicit drug use comes a time vacuum that demands to be filled as *“when someone comes into OAT, you make them redundant”* (Brené, SP). Partly, this time vacuum may be filled by the demands of treatment, such as collecting medications and attending shared care visits; however, detoxing from OAT can once again leave a gulf of time that needs to be filled. Boredom was identified as a key barrier to abstinence, with interviewees recognising this state as an important trigger for relapse to illicit opioid use. Keeping busy therefore may help individuals to maintain the structure introduced by OAT and lacking in their previous lives, as Stuart, who was post-detox, notes, “*It’s just keeping busy basically…so you go in there, got to do that…go to the laundrette, do my washing…go to the barber you know? Just random little things…yeah, normal things”.* Whilst this list of seemingly mundane activities may appear insignificant, such activities are symbolic of the normality that many service users crave, “*I want to be normal”* (Will, SU).

Another way of filling the time void is through paid employment and volunteering. Whilst some service users see these activities as a facilitator of recovery, employment may paradoxically contribute to “negative recovery capital” [64]—resources that obstruct recovery, “*It’s [employment] a help but it’s also a hinderance ‘cause it does enable me then to use*” (Keith, SU). The stability required for continuous employment may also be a barrier to reducing OAT medications and eventually leaving treatment for those for whom detox is a goal. For some, moving away from the stability that a script offers is, understandably, difficult to contemplate given that detox (particularly from methadone) is associated with a range of symptoms including pain, fatigue and sleep disruption:


*I had a client who worked for a window-fitting company, he was a lovely bloke, kept his job and everything, but it was impossible to get him off the script…because then you get to the point where the job then becomes a barrier to reducing, ‘If I reduce I’ll be in withdrawal, and I can’t be in withdrawal whilst I’m at work’. (Gabor, SP).*


Additionally, service users that want to work face barriers to doing so because of the restrictions associated with treatment. For instance, some pharmacies appear to implement policies that limit access for OAT service users to particular times, which constrain opportunities for working: 


*They [pharmacy] discourage you from going before half past eight even though they open at eight which, if you had a job or something, is like [laugh] and they are open late but you aren’t allowed to go late (Peter, SU).*


Whilst this may be due to the additional time required to prepare medications such as methadone, implicit in Peter’s comment above is the belief that this is due to discrimination, demonstrating the way that experiences of stigma in the past (enacted stigma) have resulted in Peter anticipating stigma in the pharmacy.

#### 3.3.2. Connecting with Others: Filling the Social Void

Relationships were spoken of as being both supportive and stressful, with friends that are still using being seen as a risky influence for those trying to control their drug use. This results in people severing ties with these influences, potentially leaving them socially isolated:


*…it has been really hard. It’s a bit depressing but it’s the only way I can find [blocking friends’ phone numbers] […] I’ve got a car so maybe they need a lift, […] they can tug on my heart strings and then I give them a lift and then it turns out that they’re scoring and not getting their meds [Peter, SU].*


Peter’s experiences of severing relationships with negative influences mean that for him, isolation is used as a form of self-protection. This means that isolation is a double-edged sword—simultaneously damaging and protective. It may also be one of the few ways that service users are able to exert their limited autonomy, “*It’s my choice to isolate myself ‘cause I don’t wanna get involved with other people and the chaos that comes around, especially with crack and everything else”* (Keith, SU).

Severing ties with drug-using networks therefore leaves a social void, resulting in feelings of isolation and loneliness, as service user Ringo says, *“It gets lonely all the time man yeah, I sit in my flat all the time”*. Service providers highlighted the way that austerity has compounded issues of isolation and loneliness, with wraparound support being diminished. As service provider Eric notes: *“I may be the only person that they get to speak to on a regular basis where they are not going to be judged”.* This further demonstrates the complexity of recovery within OAT and the interaction between individual, social and economic factors.

For those service users that do have access to social support, this was viewed as being a positive influence. This included informal support from others in recovery, which kept them focused on their own recovery by challenging their drug use in a constructive way:


*…I’ve got this friend who’s in the flat below […] he’s been clean seventeen years, so he’s a good person to have around me, you know. Yeah, because he used to see me and say ‘no, I don’t want to know you today mate’, I’d say ‘why’s that?’, he’d say ‘you’re pinned mate, you’ve been using’ Yeah, made me feel bad as well! (Hugh, SU).*


This embracing of informal social support contrasts with service users’ apparent reluctance to engage with structured peer support services, which may suggest that they prefer such support to develop in a more organic way. Whilst formal peer support would appear to be an excellent way of addressing the social void, service providers reported that service users were reluctant to take up the offer of this support, “*I have talked a lot about it to a lot of clients over the years, it has not had a particularly high success rate in terms of clients engaging with it”* (Russell, SP). The majority of service users had no experience of utilising this service; however, Andy described not feeling able to trust someone that had simply been through treatment themselves and had limited training, preferring instead to speak to someone with more qualifications “*I want someone with letters after their name, with a degree, because I know they put the work in, not someone that’s done a six-week psychology course”* (Andy, SU). Andy’s view was supported by the service provider Katherine, who notes that *“quite a lot of peers relapse”*. Whilst initiatives such as Mutual Aid Facilitation have been shown to increase engagement with external recovery organisations [65], service providers we spoke with said they had limited success with this approach in the past.

## 4. Discussion

We developed three themes which reflect experiences of retention, completion and recovery in OAT from the perspective of the service users and providers in our study. We found that chronic reductions to treatment budgets meant that staff felt under increasing pressure—working with people who present with more complex treatment presentations—alongside a reduction in time available to spend with service users. Service users felt disempowered and disenfranchised, lacking autonomy over important aspects of treatment, such as duration or collection of medications. Possessing overlapping identities of “injecting drug user”, “opioid user” and “OAT service user”, these individuals are vulnerable to stigma from multiple sources. Pharmacies were identified by participants as being a key site where service users experience stigma. Conversely, when interactions with pharmacists were positive, this counterbalanced experiences of stigma by increasing self-esteem and offering an important source of social support in OAT. Recovery for many is dependent on “filling the void” that heroin leaves behind; this is largely achieved through finding a sense of purpose and by reconnecting with others. However, the latter is particularly challenging for those for whom isolation is a form of self-protection. 

Together, our findings suggest that current definitions of recovery as outlined in UK policy documents [17,49] fail to appreciate the broader social, economic and relational influences on retention, completion and recovery in OAT. Here, we have shown that political, community, organisational, inter-personal and individual factors interact to determine treatment outcomes and decisions in OAT. For instance, policies which conflate abstinence with recovery have constrained treatment providers by placing increasing pressure on staff to achieve “completion targets”, whilst chronic underfunding has resulted in a reduction in staffing levels, an increase in caseloads and less time to deliver psychosocial aspects of treatment. We have shown that this directly impacts on individual treatment decisions, such as those around tapering of medications, and negatively impacts therapeutic relationships and trust between service providers and people who use OAT.

Our findings are consistent with previous studies that described the presence of extensive stigma towards people who use OAT [28,29,31,66,67,68,69]. Experiencing such stigma is believed to be a barrier to both engagement and retention in OAT [27,31,70,71,72]. Our analysis suggests that this may be because stigma is a cross-cutting influence, present at every level of the socioecological system. For instance, policies that criminalise drug use and label it as “unacceptable” [49] and define success as simply completion of treatment, may perpetuate stigma by invalidating medication-assisted recovery (long-term maintenance with OAT medications). Stigma relating to OAT has been shown to be related but distinct from that which is enacted in relation to drug use [35]. People who use OAT are therefore likely to experience stigma in relation to both their drug use as well as the treatment—so-called “*intervention stigma*” [73]. People who use OAT may also experience stigma in relation to their health status, for instance, if they are HIV-positive [74] or because they are female [75]. The impact of these overlapping, multiple stigmas, or intersectional stigma, is likely multiplicative [76]. New technologies, for instance, extended-release buprenorphine, may increase autonomy and feelings of normalcy for people who use OAT by reducing the need to visit pharmacies frequently [30]. Similarly, the COVID-19 pandemic drove the need for a revision to supervised consumption arrangements in some areas, thereby reducing exposure to stigmatising experiences in pharmacies [77].

Our findings suggest that people who use OAT feel disempowered and disenfranchised, lacking autonomy over important aspects of treatment such as duration or collection of medications. Facilitating autonomy in people who use OAT therefore is vital, not only as it is associated with increased motivation and the capacity for behaviour change [78] but because powerlessness provides the ideal conditions for stigma to thrive [33].

We identified two key voids experienced by people who use OAT—time and social connectedness. Social support is important in buffering the effects of stigma; however, filling the social void can be particularly challenging for people who use OAT as, similar to previous researchers, we found that isolation is used as a form of self-protection [37]. Remaining part of a network of individuals who continue to use drugs can facilitate feelings of community attachment; however, it may lead to increased drug use and practices such as the sharing of equipment [79]. People who use OAT may therefore cut themselves off from drug-using networks, yet stigma presents a barrier to developing more healthy support networks [80,81]. One mechanism by which this occurs may be a negative self-image as a result of internalisation of stigma. An expansion of peer support services within the drug treatment sector has been recommended to address this void [48]; however, whilst such initiatives may be a key way of addressing social isolation, our participants identified barriers to accessing such support, and future research should seek to understand these barriers in more depth.

### 4.1. Implications for Policy and Practice

Many of the issues apparent from our data appear to stem directly from the chronic funding reductions to drug and alcohol treatment services in the UK in recent years, which have resulted in a de-skilling of the workforce and reduced capacity [22,48,82]. Alongside key organisations in the drug treatment field (e.g., [83]), we support the assertion of Dame Carol Black that addiction treatment in the UK requires “*radical reform*” [48]. The UK government’s Project ADDER [84] is providing GBP 59 million to local authorities in eight areas of high drug-use prevalence to enhance treatment and recovery provision, including Bristol. These findings were recently presented to key stakeholders leading ADDER in Bristol. By detailing specific issues with the current “*broken system*”, we have highlighted areas for reform. Whilst funding increases are welcome, the benefits of this may be limited by continuing to focus on and incentivise treatment completion over retention [82]. Our findings, therefore, will be of particular interest for policy makers and service leads who wish to effectively allocate funding to specifically improve the provision of OAT. For instance, we have supported the findings of others that delivery of psychosocial aspects of treatment may be compromised by larger and more complex caseloads [85].

### 4.2. Implications for Intervention Development

In line with previous research, we found that stigmatising views towards people who use OAT may be enacted by individuals who work in healthcare settings, including GP surgeries and pharmacies [30,73,86,87,88]. This may be driven by a preference for abstinence-based treatment as well as poor knowledge of OAT among healthcare professionals [73]. Given the established link between stigma and treatment retention, efforts should be made to develop interventions to reduce all types of stigmas (enacted, anticipated, internalised) in relation to OAT. Such interventions should aim to increase knowledge of OAT and challenge the perception of recovery as abstinence alone. By considering our findings in the context of the socioecological model [38], future intervention developers may wish to consider the presence of stigma at each level of the system and the way stigma mechanisms interact. For instance, to reduce internalised stigma, it is likely to be necessary to address enacted stigma in tandem [35]. Healthcare settings such as GP surgeries and community pharmacies are natural targets for the development of such interventions. Early findings from a large-scale programme commissioned by the Australian government to reduce stigma towards key groups, including people who use drugs, suggest that it is possible to effectively shift attitudes of healthcare workers [89].

### 4.3. Strengths and Limitations

By considering findings in relation to the socioecological model, we have shown that treatment journeys within OAT are constrained by complex interactions between the individual and wider contextual factors. Our study therefore builds upon the work of previous researchers who have explored treatment experiences in OAT from an ecological perspective (e.g., [39,45,46]) and specifically offers consideration of these factors within the UK treatment system. Adopting a systems perspective is key to developing interventions that take into account contextual factors and the relationships between these [90,91].

Our findings offer important insights into the concepts of retention, recovery and treatment completion within OAT from the perspective of both service users and providers. By identifying patterns across the experiences of those both receiving and providing OAT, we have ensured that future intervention development will be rooted in the experiences of service users with acknowledgement of the challenges and opportunities of delivering treatment. For instance, we have identified that staff feel under increasing pressure to carry out their duties effectively, which is something that, to our knowledge, has not been previously explored elsewhere. 

Limitations on time resources meant that we were unable to interview pharmacists or GPs within this study; however, doing so may have offered important perspectives on the provision of OAT, given that they work in close partnership with local drug treatment services. As with any qualitative study, our findings are geographically and temporally situated. The interviews took place immediately prior to the COVID-19 pandemic and a year before the UK government released a new drug strategy. The arrival of the pandemic drove global changes to the way that OAT is delivered, for instance, revisions to supervised consumption arrangements, to maintain social distancing and improvements to access to OAT in the absence of other services [92]. Whilst evaluations are ongoing, it would appear that such initiatives have the potential to increase engagement, reduce waiting times [93] and reduce experiences of stigma in the pharmacy setting [77]. However, recent research indicates that there was an increase in methadone-related mortality during the first wave of the pandemic in individuals who were not prescribed the drug [94]. Clearly, a balance needs to be sought between addressing stigma and minimising harm.

The study was carried out in Bristol, where OAT is delivered through a shared care arrangement between GP surgeries and a third-sector organisation. In other areas of the UK and elsewhere globally, OAT may be provided from within third-sector organisations alone. Readers should therefore consider the transferability of these findings to their own contexts.

### 4.4. Conclusions

By taking an ecological approach, we have considered the multi-level influences on retention, completion and recovery within OAT. Our study has presented a picture of a “broken system”, one that requires “radical reform” [48]. The presence of stigma at every level of the socioecological system presents multiple barriers to recovery and limits the life chances of those receiving OAT. Defining success in OAT as “completing” treatment whilst neglecting other important outcomes means that even those individuals that are able to detox from OAT do so in such a way that it leaves them “holding on by the seat of their pants” (Ronnie, SP). With little aftercare and wraparound support available, this leaves these individuals vulnerable to relapse and exposed to unnecessary risk. We therefore argue for consideration of relational, economic and political factors in addition to individual-level explanations of recovery from opioid use disorder.

## Figures and Tables

**Figure 1 ijerph-20-01526-f001:**
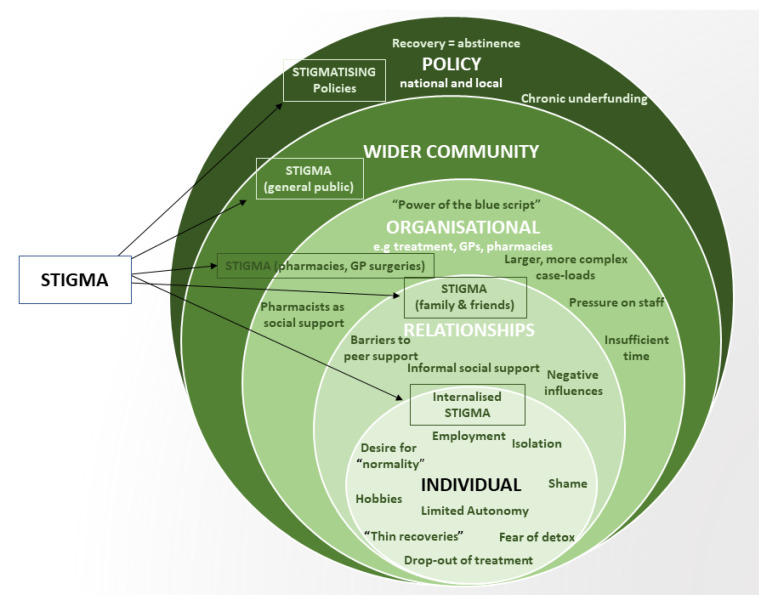
Factors influencing treatment journeys at each level of the socioecological system. Stigma is a cross-cutting influence on treatment journeys in OAT.

**Table 1 ijerph-20-01526-t001:** Service user participant characteristics (n = 12).

Pseudonym	Gender	Medication	Treatment History *	Age (Years)
Andy	Male	None (post-detox)	3 months	45
Ringo	Male	None (post-detox)	4 years	39
Robert	Male	None (post-detox)	7 months	50
Stuart	Male	None (post-detox)	18 years	42
Anne-Marie	Female	Buprenorphine	20 years	52
Hannah	Female	Buprenorphine	>20 years	56
Jack	Male	Buprenorphine	>10 years	53
Peter	Male	Buprenorphine	12 years	37
Davina	Female	Methadone	20 years	56
Hugh	Male	Methadone	>25 years	75
Keith	Male	Methadone	25 years	46
Will	Male	Methadone	13 years	41

Note: * treatment history was time since first engagement with OAT services over lifetime and not limited to contact with current treatment duration.

**Table 2 ijerph-20-01526-t002:** Service provider participant characteristics (n = 13).

	n (%)
Role	
Shared care worker	5 (38)
Shared care team leader	2 (15)
Community detox link worker	2 (15)
Clinical psychologist	1 (8)
Service manager	3 (23)
Years of experience in current/similar role	
1–5	5 (38)
6–10	3 (23)
11–20	5 (38)

Note: pseudonyms are not provided here to protect anonymity of participants due to a small number of participants and possible identification.

**Table 3 ijerph-20-01526-t003:** Indicators of recovery suggested by participants as alternatives to abstinence.

Outcome	Participant	Illustrative Extracts
**Health**		
Help-seeking for health complaints	Jenny (SP) Marc (SP) Brené (SP)	*[…] getting people to engage with Hep [hepatitis] treatment to maintain all their appointments with the [Unit for hepatitis treatment] and just address their physical health problems.* *We’re contributing to get more people into hepatitis treatment, to get more people tested…* *There’s so many more outcomes that you can be looking at, like well-being, physical health, emotional health, education, employment…* *We need to start looking at how many people have got a GP practice…*
**Individual**		
Stability	Anne-Marie (SU)	*You can’t be constantly detoxing you get a point where you’re just like ‘I need to just get on with my life and I’ll worry about it in six months’ or something’.*
	Russel (SP)	*I think being on that script has made a massive difference to his [service user’s] life. Now as a consequence he is living in a lower supported accommodation. He is much more stable.*
	Gabor (SP)	*Stability, life stability, that’s more important than being drug free to me, and I know that can mean being on a script for a lot longer…*
	Stuart (SU)	*[…] putting the foundations in place…and then the next stuff are like your hobbies and things you want you know you’re sort of building that wall higher and higher because the shits’ over there and you’re not going to be able to climb over it anymore because you’ve built up so much stuff that it’s solid enough to keep you safe on that side…*
Stable housing	Jack (SU)	*I’ve come a long way from the days of homelessness and being in the gutter with nothing not even any drugs then, you know what I mean? nothing, just some clothes.*
Self-care	Russel (SP)	*I think being on that script has made a massive difference to his life…He is looking after himself. He has got himself a bike, he is cycling every day on the bike.*
**Inter-personal**		
Wellbeing of children	Eric (SP)	*Stop defining positive recovery as drug free. I mean, we have all got loads of clients who come into us and been chaotic and made slow changes and you know, now the kids are being fed […]. Fucking amazing, that alone is positive.*
	Russel (SP)	*I had a client today that came in. Him and his partner have been using on and off for years. They have got three children. Social services are involved. Children are on the child protection register. He came in today buzzing saying to me, I haven’t used for three weeks…and he said to me, the children are happy. I said, well, mum and dad are happy, children are probably more likely to be happy. If he didn’t have that script where would he be? It is life changing. I can’t emphasise enough really how I think it is crucial that people that need OST [OAT] get OST.*
Healthy support networks	Russel (SP)	*We need to start looking at…how many people have got positive social relationships outside of the drug user networks*
Relationships	Russel (SP)	*He [service user] didn’t have a good relationship with his mother; he suffered physical abuse and neglect.…emotional abuse…He actually said to me, when we were coming to the end of his treatment that because of what he had experienced he didn’t trust women and found it really difficult. And, actually, after working with me over a long period of time, he felt he could trust women now. The person that took over from me was another female and he had a good relationship with her.*
	Russel (SP)	*I think being on that script has made a massive difference to his life…He has talked about regaining contact with his adult daughter. They had a good relationship, but he said they had not seen each other for a couple of years. He went to prison and he has not been in touch with her since…I saw him yesterday and I thought, wow, you’re a different person than I met the first time*
Domestic violence	Jenny (SP)	*The only thing that’s counted is the drug free stat, all the little steps that get you there aren’t really… when it comes to my supervision—how many have you got drug free? Any drug frees coming in the next month? It’s sort of like, well no really, but then all this work I’ve done with this client who’s having horrific domestic violence…what about that? Is that not counted? […] I was worried that I was going to come into surgery and find out that she’d been murdered or something, whereas now it feels like the early days of recovery for her.*
**Financial**		
Access to benefits	Brené (SP)	*[…] we need to start looking at how many people have been linked up with housing services, how many people have been in touch with benefits advice…*
Access to education and employment	Gabor (SP)	*there’s so many more outcomes that you can be looking at, like well-being, physical health, emotional health, education, employment…*
**General**		
Medically Assisted Recovery (MAR) = successful treatment	Brené (SP)	*I don’t see why there can’t be an exit code to say actually successful completion, medically assisted recovery and we have recovery check-ins with them.*

## Data Availability

Data sharing not applicable. The data generated for this study is qualitative in nature and not available due to ethical restrictions.

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
