# Peer review of "Should I Stay or Should I Go? A Qualitative Exploration of Stigma and Other Factors Influencing Opioid Agonist Treatment Journeys"

_ijerph, 2023, doi:10.3390/ijerph20021526_

Round 1

Reviewer 1 Report

IJERPH

This is a well written contribution to the literature on barriers to OAT retention. Using a socioecological framework to highlight systemic issues is a strength of this paper. Additionally, findings on filling the void (i.e. time) and stigma experienced in pharmacy settings are important contributions to the existing literature. I have made some minor suggestions to help strengthen the final version of this manuscript.

Introduction

Well written, organized, and cited.

You make an important point about the differing vulnerabilities in early treatment between methadone and buprenorphine. It would benefit this section to make this point as a distinct sentence versus a parenthetical aside. Your sample of OAT service users include those taking methadone, so this additional information would be valuable.

Can you add more here about the Community Discharge Link workers and general process? This will provide important context for the methods. For example- at what point are people encouraged to detox from OAT medications? What indicators are used to determine this is the next step in treatment? Etc. This is different from the US (my context) so I am interested in learning more about this.

Given that stigma is a central part of your framework, the introduction would benefit from presenting some of this existing literature. I would suggest moving the framework of stigma from the results section to the introduction section.

Similarly, some discussion of UK drug policies would be useful context for the introduction.

Methods

I appreciate your discussion of reflexivity. Can you elaborate on the person who helped develop recruitment materials? An additional sentence would help here. Additionally, a few brief sentences on the positionality of the research team would strengthen this section.

Please define shared care workers (ln. 131).

In your discussion of information power, please clarify that this concept applies to qualitative methods.

Results

Do you have any additional data on service user participants? For example, age or race/ethnicity?

Please clarify if you used pseudonyms for the service users.

Not all quotes are presented in italics. Please edit for consistency.

Table 3 presents extremely important information. It does not make sense to me that this is embedded within the “system is broken” theme. I would suggest making this table an additional and distinct thematic finding (i.e., indicators of recovery). Alternatively, you could potentially make this a sub-theme of the ‘system is broken’ theme.

In the filling the void section you talk about time in terms of filling time previously filled by seeking and using drugs. You also discuss time in terms of access to prescriptions. These are quite different and equally important aspects of time. As such, they might benefit from being separate sub-themes.

Discussion

I would suggest clarifying that “the pandemic” you are referring to is COVID-19. People reading this in the future may need that clarifying context (ideally not, but these are unpredictable times!).

Author Response

Thank you for taking the time to review our manuscript and for your constructive suggestions and comments, which we hope we have adequately addressed (in blue) in the attached file. 

Reviewer 2 Report

In the manuscript entitled " Should I Stay or Should I Go? A qualitative exploration of 2 stigma and other factors influencing opioid agonist treatment journeys. The authors sorted out opioid agonist treatment and stigma related to treatment, the interaction between service takes, service providers, pharmacists, and broken system, and how OAT treatment outcomes are influenced by socioecological characteristics.

They also gave recommendations to solve the broken system.

Minor comments:

Nil.

Major comments:

1)      Section 2.2 is there any inclusion and exclusion criteria for the selection of OAT.

2)      Authors have only 4 subjects for each treatment for methadone and buprenorphine which have differences in PK/PD activity, with only 4 subjects in each drug did you get statistical significance. PMC3271614

3)      Table-1 post-detox group took which medication.

4)      What is the route cause of subjects thinking the use of OAT is a badge of shame.

5)      The mental health issues were majorly due to free treatment sponsored by the government are there similar issues with subjects paying for their own treatment?

6)      What are the limitations of the current study can be added.

Reviewer 3 Report

Dear authors,

Congratulations for the extensive work you have performed and for the topic of interest, which I consider of the most relevance. It was a pleasure to read it and I have no doubts that it will be of great importance to policymakers and Public Health. 

  • This is a qualitative study aiming to understand the way different socioecological aspects interact to influence treatment outcomes in opioid agonist treatment;
  • It is an original topic contributing to the knowledge in the field. It addresses a gap in the literature and the authors, supported in their findings, presented areas for reform in a local project. This is a good example of how qualitative findings can be useful and must be transferred to the field, contributing to the quality of life of a vulnerable and stigmatized population;
  • This study offers insights into the concepts of retention, recovery and treatment completion within opioid agonist treatment from the perspective of both service users and providers, which adds value to the practice, policymakers and Public Health;
  • Multiple factors that can influence the experience of the treatment are explored which improves the accuracy of the findings;
  • The methodology is sound. Data collection was adequate and the saturation is well explained. Data analysis is detailed and adequate;
  • Conclusions are consistent with the evidence and answer the study purpose;
  • References are appropriate and recent (56% have less than five years);
  • Figure and tables are clear, understandable and readable;
  • It is well written.

On page 11, correct "autonomy" in your statement "One suggested method of increasing autotomy might be by allowing for less frequent appointments for those in long-term..."

I have no other recommendations for improvement. Congratulations once again and wish you all the success!

Author Response

Thank you for taking the time to review our manuscript and for your positive feedback, which we very much appreciate. Thank you for spotting the typo, which we have now corrected.